# Biochemical recovery from exertional heat stroke follows a 16-day time course

Matthew D. Ward[1], Michelle A. King[1], Charles Gabrial[2], Robert W. Kenefick[1], Lisa R. Leon[1]*

1 Thermal and Mountain Medicine Division, United States Army Research Institute of Environmental Medicine, Natick, Massachusetts, United States of America, 2 Defense Health Agency, Falls Church, Virginia, United States of America

* lisa.r.leon.civ@mail.mil

**Data Availability Statement:** All relevant data are within the manuscript and its Supporting Information files.

**Funding:** This work was supported by the United State Department of Defense, Defense Health

## Abstract

### Background

The aim of this study was to characterize the time-resolved progression of clinical laboratory disturbances days-following an exertional heat stroke (EHS). Currently, normalization of organ injury clinical biomarker values is the primary indicator of EHS recovery. However, an archetypical biochemical recovery profile following EHS has not been established.

### Methods

We performed a retrospective analysis of EHS patient records in US military personnel from 2008–2014 using the Military Health System Data Repository (MDR). We focused on commonly reported clinical laboratory analytes measured on the day of injury and all proceeding follow-up visits.

### Results

Over the prescribed period, there were 2,529 EHS episodes treated at 250 unique treatment locations. Laboratory results, including a standardized set of blood, serum and urine assays, were analyzed from 0–340 days following the initial injury. Indicators of acute kidney injury, including serum electrolyte disturbances and abnormal urinalysis findings, were most prevalent on the day of the injury but normalized within 24-48hours (creatinine, blood urea nitrogen, and blood and protein in urine). Muscle damage and liver function-associated markers peaked 0–4 days after injury and persisted outside their respective reference ranges for 2–16 days (alanine aminotransferase, aspartate aminotransferase, creatine phosphokinase, myoglobin, prothrombin time).

### Conclusion

Biochemical recovery from EHS spans a 16-day time course, and markers of end-organ damage exhibit distinct patterns over this period. This analysis underscores the prognostic value of each clinical laboratory analyte and will assist in evaluating EHS patient presentation, injury severity and physiological recovery.

Program, grant number W81XWH-13-MOMJPC5-IPPEHA (authored by LRL; https://cdmrp.army.mil/dmrdp/default). The funders had no role in study design, data collection and analysis, decision to publish, or preparation of the manuscript

**Competing interests:** The authors have declared that no competing interests exist.

## Introduction

Currently, considerations for recovery and return to activity decisions after exertional heat stroke (EHS) are highly subjective and circumstantial (based on 'best guess' [1]), despite extensive injury characterization in day laborers [2–4], religious pilgrims (Hajjis) [5–7], athletes [8–10], military personnel [11–16], and experimental animal models [17–19]. Previous studies have failed to build a consensus or generalizable guideline for normalization of clinical laboratory markers following EHS, even though these markers are currently used as the primary indicator of EHS recovery by the American College of Sport Medicine (ACSM), the National Athletic Trainers' Association (NATA), and the United States Army [1,20–22].

End-organ injuries are among the defining features of EHS, although initial presentation and diagnosis is established by central nervous system dysfunction with severe hyperthermia [23]. Experimental models of EHS, and autopsies from fatal EHS cases have shown that nearly every organ is impacted [11,19]. This organ and tissue injury is a consequence of the cytotoxic effects of heat, and hypoperfusion of splanchnic tissues due to increased peripheral blood flow [24]. The magnitude of organ injury and mortality rates are thought to be correlated with the thermal load, a function of the degree and duration of hyperthermia [17,18].

Evidence of acute kidney injury, hepatic insufficiency or liver damage, myolysis, and disseminated intravascular coagulation are apparent in nearly all EHS patients. In most cases, these symptoms are transient and often completely reversible [2,3,8]. In rare fatal cases of EHS, patient prognosis and organ injury indicators will continue to worsen within days of the precipitating injury. Patients who succumb to EHS experience severe disturbances in clotting mechanisms, cerebral hemorrhage, and organ failure [4,11]. In non-fatal cases, the time course of biochemical recovery has not been well-described. An improved understanding of EHS pathophysiology, etiology and the factors that influence recovery will improve decision making regarding treatment, and return to activity [25].

In order to more completely describe EHS recovery, we accessed electronic health records (EHR) in the US Military Health System (MHS). The MHS is a worldwide healthcare system with standardized, and centrally-managed EHRs. The US Armed Forces also recommend a standardized subset of clinical laboratory analyses, as part of heat casualty management treatment guidance (summarized in S1 Table) [23,26,27]. These uniform records and defined medical practice provide a unique opportunity to access patient data that are otherwise inaccessible in comparable civilian healthcare domains.

The purpose of this study is to better inform EHS patient management by describing objective measures of physiological recovery. Defining an archetypic biochemical recovery profile in EHS patients enables evidence-based recovery evaluation, and will further help medical providers access the right laboratory analysis, at the right time, for the right reason [28].

## Methods

We identified a retrospective cohort of EHS patients using the Military Health System Data Repository (MDR), an EHR management system employed by the MHS. The study was limited to US Active Duty Service Members (US Army, Navy, Air Force, and Marine Corps) serving from 2008–2014 who received a diagnosis of heat stroke, code 992.0 of the International Classification of Disease, Ninth Revision (ICD-9). It is important to note that an ICD-9 code distinguishing EHS and classic heat stroke does not exist; heat strokes in the military are presumptively considered exertional, absent information about injury etiology. This conjecture is supported by previous studies showing that >90% of reported heat strokes among US military populations injuries occurred during intensive training activities or physical exercise [11,12,14,16].

Only pre-determined variables were drawn from de-identified EHRs in accordance with the minimum necessary standards of US federal law for patient privacy protections. This analysis included patient demographic variables including age, gender, body mass index, and race, and variables associated with patient care including clinical laboratory values, follow-up care information, and total treatment costs. Total treatment cost were calculated as the cumulative cost to the health care system including initial care and all follow-up or in-patient care associated with the initial injury. Laboratory measurements were performed at each treatment location using equipment and procedures as indicated by local medical practice. The study was reviewed by the United States Army Research Institute of Environmental Medicine and the United States Army Medical Research and Materiel Command institutional review boards and determined this research exempt from written informed consent requirements.

### Defining features of episodes and follow-ups

An episode of EHS included the initial clinical encounter (heat stroke, 992.0) and all available follow-up visits. Subsequent clinic visits were identified as follow-ups if the visit occurred within 60-days of the initial encounter or most recent follow-up. Visits occurring greater than 60-days from the previous follow-up were considered an initial encounter for a subsequent episode. The 60-day washout period to delineate episodes was based on personal communication with a uniformed physician familiar with EHS follow-up care in the military system, as well as broad risk-based considerations for planning appropriate and timely medical follow-up [29].

### Considerations for clinical laboratory result analyses

For analytes drawn multiple times on a particular day for a patient, the repeat measures were averaged and reported as a single value for that day. To orient the analysis of laboratory values around the upper limits of normal (ULN) and lower limits of normal (LLN), we aggregated all reported reference ranges and identified the most frequently reported ranges as 'consensus' reference ranges.

For normally distributed continuous variables, laboratory results were presented as mean ± standard deviation. Non-normally distributed variables were presented as median and interquartile range. Fold changes were calculated using population median or mean, relative to the ULN or LLN.

Urinalysis results were non-uniformly reported as both continuous (numerical) or categorical (free text) variables (i.e., urine myoglobin as '10 ng/dL' or 'Positive'). In these instances, the continuous variable was rendered into a categorical equivalent, as either 'normal' or 'abnormal'. Data set characterization and descriptive statistics were generated using GraphPad Prism 7 (GraphPad Software, Inc; La Jolla, CA).

## Results

### Patient characteristics, available laboratory results

Over the study period, we recorded 2,216 patients experiencing a total of 2,529 heat stroke episodes (Table 1). These diagnoses originated from 250 distinct treatment locations, at 43 international and 207 domestic US sites. On average, patients received 2.6 follow-up visits, over an 18-day period following EHS. The sum of treatment cost for these episodes was $13,497,524 over the study period (average, $5,337 per patient-episode).

Sixty percent of patients reported at least one clinical laboratory value in the EHRs on the day of the initial injury (Fig 1 and S1 Table). Reported laboratory data decreased precipitously

**Table 1. Episode, treatment and patient descriptors.**

| | | | |
|---|---|---|---|
| **TOTAL EPISODES (ICD-9: 992.0)** | | **2,529** | |
| *Episode descriptors* | | *Average (Range)* | |
| | Average episode length (Range) † | 18 (0–340) | |
| | Average no. follow-ups visits (Range) | 2.6 (0–58) | |
| Average cost per episode | | $5,337 ($0–497,293) | |
| Total treatment costs (sum, all episodes) †† | | $13,497,524 | |
| **TOTAL TREATMENT LOCATIONS** | | **250** | |
| | Domestic | 207 | |
| | International, or 'at sea' | 43 | |
| **TOTAL PATIENTS** | | **2,216** | |
| No. of patients with lab results in EHR | | 1,675 | |
| *Patient Demographics* | | *Frequency* | *Percent* |
| **Gender** | Male | 1,997 | 90.1 |
| | Female | 219 | 9.9 |
| **Race** | White | 1,081 | 59.3 |
| | Black | 228 | 12.5 |
| | Asian or Pacific Islander | 80 | 4.4 |
| | American Indian/Alaskan Native | 11 | 0.6 |
| | Other | 375 | 20.6 |
| | Unknown / Multi-racial | 48 | 2.6 |
| | *Not indicated* | 393 | – |
| **Age** | <20 | 322 | 14.5 |
| | 20–24 | 907 | 40.9 |
| | 25–29 | 488 | 22.0 |
| | 30–34 | 250 | 11.3 |
| | 35–39 | 140 | 6.3 |
| | 40+ | 109 | 4.9 |
| **BMI** | <18 | 7 | 0.5 |
| | 18–26 | 933 | 60.0 |
| | >26 | 614 | 39.5 |
| | *Not inidicated* | 662 | – |

†Episode length = number of days between injury and last coded follow-up

††Based on associated billing codes for all episodes, initial treatment and all follow-ups

EHR = electronic health record

on subsequent days, with less than 10% of the cohort reporting laboratory results four days post-injury.

## Liver dysfunction, myolysis markers in EHS patients

Acute liver dysfunction and myolysis are evident in nearly all patients with available laboratory results. Select serum enzymes, including alanine amino transferase (ALT), aspartate amino transferase (AST), creatine phosphokinase (CK) and myoglobin (Myo), peak and clear from circulation in distinct patterns through days 0–14 (Fig 2A). Median values for other liver function enzymes including lactate dehydrogenase (LDH), alkaline phosphatase (ALP), bilirubin (BILI) and albumin (ALB) were within their respective reference ranges on the day of the injury and all follow-up days (S2 Table).

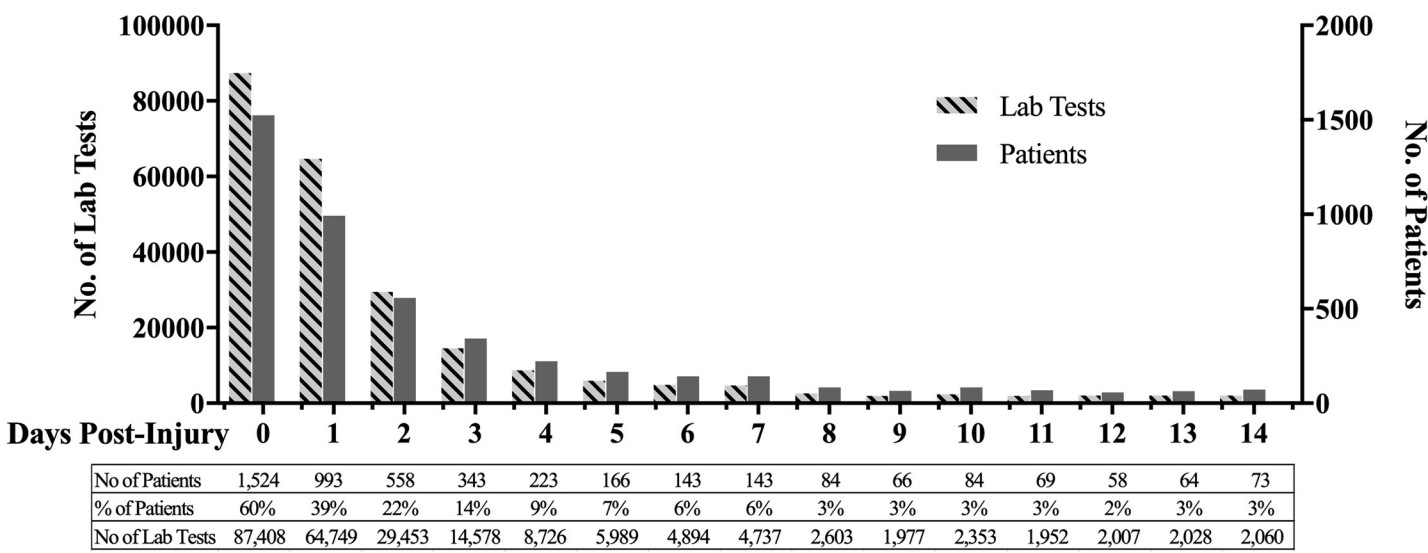

**Fig 1. Frequency of laboratory analysis days following EHS.**

### Acute kidney injury, serum electrolytes and abnormal urinalysis in EHS patients

Serum electrolyte values indicating acute kidney injury generally resolve within 24-48hrs [10,16]. Creatinine (Cr) was elevated above the ULN in 58% of patients, and urea nitrogen (BUN) was elevated in 34% of patients on the day of the injury (day 0, Fig 2B and S3 Table). Glucose, potassium, and sodium were within reference range on average, however, appreciable sub-populations of above the ULN and below the LLN are apparent on day 0. Previously unappreciated disturbances in calcium (Ca) and chloride (Cl) values are reported 1-day following injury; 68% of patients presented hypocalcemic, and 54% of patients exhibited mild hyperchloremia (Fig 2C, S3 Table).

The prevalence of abnormal urinalysis results were a defining feature of this cohort, including elevated urine protein, blood, ketones and myoglobin on the day of the injury (S4 Table). Abnormalities were most prevalent on day 0, and exhibited variable prevalence and persistence in the population on proceeding days.

### Coagulation and blood counts in EHS patients

Disseminated intravascular coagulation (DIC) is a commonly annotated feature of EHS [8,12]. In this population, prothrombin times (PTT) peaked 4 days after injury and persisted outside the reference range from days 1–16 days (Fig 2C). Corresponding decreases in hemoglobin (Hgb), hematocrit (HCT) and red blood cells (RBC) reached nadir 1 day after injury, and persisted below the LLN for 6 days (Fig 2C and S5 Table). Platelet counts for 35% of patients were below the LLN on day 2 (thrombocytopenic; S5 Table), however, the population mean remained within the reference range. The mean white blood cell (WBC) and neutrophil (Neu) counts were elevated above the ULN only on day 0, indicating a transient elevation of immune activity immediately following injury.

### Summary, biomarker magnitude and persistence

Overall, disturbances in clinical laboratory analytes peak between 0 and 4 days following injury, and persist outside their respective reference ranges for up to 16 days (Table 2). The

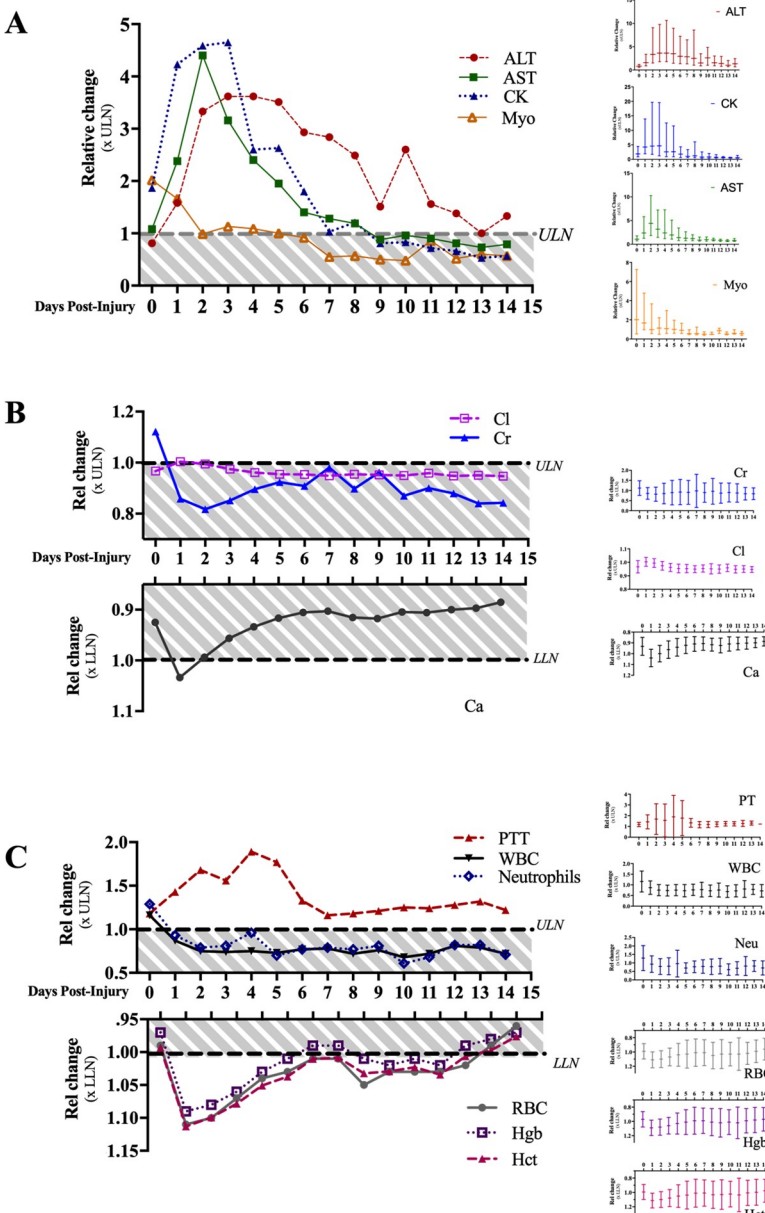

**Fig 2. Profile of biomarker changes days following EHS.** Biomarkers of multi-organ injury are represented as fold-change relative to limits of reference ranges; ULN = Upper Limit of Normal, LLN = Lower Limit of Normal. *(A)* Liver function and myolysis-associated enzymes shown as median values; in-set graphs show inner quartile range for each analyte. (*B*) Serum electrolytes, and (*C*) hematology and coagulation results shown as mean; in-set graphs show standard deviation for each analyte and day. Shaded values indicate values within reference range.

incidence and magnitude of these extreme values varies from immediate and slightly depressed (RBCs; Day 1, 1.12 xLLN fold-change) to delayed and significantly elevated (CK; Day 3, 4.65 xULN fold-change). Most disturbances persist for only a single day, (Cr, Ca, Cl, Myo, WBC and Neu). Persistent disturbances in laboratory analytes, including PTT, AST, ALT, and CK, peak later and recover more slowly, and define the protracted nature of bio-chemical recovery.

**Table 2. Summary of biomarker disturbances, magnitude and persistence.**

| VARIABLE | MAGNITUDE | | PERSISTENCE |
|---|---|---|---|
| | Peak/Nadir (Day) | Amp of Peak (xULN/LLN) | Outside Ref Range (Days) |
| **HEMATOLOGY, COAGULATION**[†] | | | |
| Red Blood Cell | 1 | 1.12 x LLN | 1–12 |
| Hemoglobin | 1 | 1.09 x LLN | 1–11 |
| Hematocrit | 1 | 1.11 x LLN | 1–12 |
| White Blood Cells | 0 | 1.15 x ULN | 0 |
| Neutrophils | 0 | 1.001 x ULN | 0 |
| Prothrombin Time | 4 | 1.89 x ULN | 0–16 |
| **ELECTROLYES, RENAL FUNCTION**[†] | | | |
| Creatinine | 0 | 1.12 x ULN | 0 |
| Calcium | 1 | 1.03 x LLN | 1 |
| Chloride | 1 | 1.003 x ULN | 1 |
| **LIVER FUNCTION, MUSCLE INJURY**[††] | | | |
| Alanine Aminotransferase | 3/4 | 3.62 x ULN | 1–14 |
| Aspartate Aminotransferase | 2 | 4.41 x ULN | 0–8 |
| Creatine Kinase | 3 | 4.65 x ULN | 0–8 |
| Myoglobin | 0 | 2.01 x ULN | 0 |

†Based on population mean value

††Based on population median value

## Discussion

Here we present the largest and most comprehensive retrospective characterization of EHS cases to-date. These findings fill a critical gap in understanding the time-resolved progression of biochemical recovery, days following EHS injury. This analysis will serve as a guide to better understand the diagnostic and prognostic significance of laboratory values from the point of EHS injury throughout biochemical recovery. We also defined 16-days as an anticipated time-frame of laboratory value normalization, the first recovery milestone and foundation for return to activity decisions following injury [1,20].

### Similar studies and analysis of EHS presentation and recovery

Previous studies investigating the time course of biochemical recovery concluded that the extent of EHS multi-organ injury is highly individualized and recovery rates are unique to each patient [14,20]. Our findings challenge this conclusion, as the consensus biochemical recovery profile presented here is coherent, predictable and uniform, albeit varying in degree.

Previous clinical descriptions of EHS presentation and recovery have limited generalizability, due to small patient populations [12,30,31], and often focus on the most severe or fatal forms of injury [11,32]. This study includes a much larger cohort, and includes all available cases without regard for an arbitrary severity designation. Our study further surveys patient variables from 250 unique treatment locations across the world, and more broadly describes heat casualty management instead of the practices and medical resource availability of a single treatment location or region [10,16,33]. The key strength of this study is the availability of laboratory follow-up days after the injury. Previous reports focus on physiological disturbances at the point of injury, and highlight patient management during initial treatment, and seldom emphasize the longitudinal significance of the same disturbances on proceeding days. As a

result, current recovery assessment is highly individualized, based on circumstantial improvement of a patient.

## Interpreting clinical laboratory values, and implications for treatment and return to activity

For clinical laboratory values, timing is critical. Following EHS, indicators of kidney injury (Cr, BUN) appear early, and resolve quickly. Conversely, indicators of liver impairment (i.e., PTT, ALT, AST, CK) peak days after the injury, and normalization is protracted. Therefore, markers of liver damage and dysfunction are better indicators of complete biochemical recovery. It is important to note that the source of circulating enzymes indicative of liver dysfunction are likely the combined outcome of both exercise-associated myolysis and hepatocellular injury or insufficiency.

We identified two previously unappreciated findings related to hyperchloremia and hypocalcemia one day following EHS injury. Elevated chloride values are likely benign, and a direct result of extensive intravenous normal saline use during the course of immediate EHS treatment [34]. It is unclear from this study if hyperchloremia findings are clinically actionable. However, it has been recommended in rhabdomyolyis-associated fluid resuscitation to alternate between use of normal saline ([$Cl^-$] = 154 mEq/L) with lactated ringers ([$Cl^-$] = 109 mEq/L) [35,36]. Decreases in calcium have also been previously demonstrated in rhabdomyolysis [37], attributed to calcium deposition in damaged muscle tissue. More evidence is needed to determine the implications of these electrolyte disturbances in EHS, as compared to rhabdomyolysis.

We also highlighted the apparent discrepancy in prolonged abnormal findings in urinalysis as compared to corresponding serum electrolytes indicative of renal insufficiency or acute kidney injury. Our analysis indicates that urinalysis may be more indicative of persistent kidney damage or dysfunction, due to the prevalence of abnormalities among patients in this cohort.

Consensus recommendations for return to activity by the NATA, ACSM and US Army identify either 7 or 14 days as the minimum post-incident rest period before starting a gradual return to activity or training[1,21,22]. Each of these consensus recommendations also indicate 'normal blood-work results' and/or medical clearance from a physician's evaluation as a more individualized indicator of medical readiness. The duration of biochemical recovery described in this analysis strongly supports a minimum 14-day recovery period. However, based on the observed variability in biochemical recovery patterns among individuals, a 7-day post-incident physical examination including laboratory analysis would help identify less severe presentation of EHS, and facilitate an earlier return to activity."

## Frequency of laboratory analysis, and treatment cost

One of the unanticipated findings in this study included a frequency survey of follow-up visits and laboratory analyses (Tables 1 and 2). We conjecture that lower than expected laboratory data completeness may be due to (i) the EHS standard-of-care due to limited medical resources availability, or (ii) the positive prognosis of patients immediately, and days-following initial treatment, who may have required minimal medical observation, laboratory analysis, and follow-up (Tables 1 and 2). This is in agreement with previous findings [38,39], that contemporaneous emphasis on injury prevention, and immediacy of EHS treatment has improved patient prognosis and made EHS a highly recoverable event. Limited medical documentation and availability of laboratory values on the day of the injury reflects similar standards of care available at mass participation athletic events or road-races [39].

We also summarized the cost burden for EHS treatment. This value is a dramatic underestimate of the total cost of EHS, and does not account for lost work or duty days, and the long-term health consequences, such as the increased subsequent hospitalization rates documented for afflicted individuals [40]. This is the best estimate of treatment cost currently available.

### Weaknesses, opportunities for future investigation

One of the key constraints of this retrospective study is the limited apparent availability of laboratory values on days following injury, circumstantially dictated by the practical considerations of attending physicians and emergency medical providers. We conjecture that the apparent low data completeness, especially on days following injury, may bias our analysis towards more severe patients and those requiring more extensive medical oversight. An unbiased prospective study with systematic laboratory analyses would clarify this proposed bias.

While we have focused on the longitudinal nature of biochemical recovery, it is important to note that the initial clinical laboratory evaluation alone plays a vital role in early medical management of suspected EHS patients. For example, electrolyte abnormalities including hyponatremia will dictate intravenous administration of normal saline, and other treatment and medical decision-making.

Previous case reports and case series have shown that patient prognosis and biochemical recovery are largely influenced by initial treatment variables and modalities, most notable including the immediacy of aggressive cooling treatment [31,33]. Our source data lacks the detail and resolution needed for this type of analysis. It is our hope that defining an archetypical biochemical recovery profile in a large cohort will facilitate comparison of other treatment modalities and practices between medical treatment facilities.

Pathological laboratory disturbances following EHS are marginally distinguishable from metabolic changes following exhaustive exercise [18,41,42]. Also, attempts to verify tissue-specific enzyme release patterns to increase specificity of organ injury have not been adopted in the clinical setting [6,18]. More sensitive and specific laboratory markers of EHS injury are needed to differentiate between exercise-associated stress and fulminant EHS injury.

### Conclusion

This retrospective analysis is the first large-scale, multi-site study to describe a time-resolved recovery profile of clinical laboratory values following EHS. It further defines the standard recovery time for biomarker normalization at 16-days. The distinct pattern of biomarker disturbances indicate the prognostic value of each analyte immediately and days after injury. These findings will guide recovery and return-to-training decisions and have broad implications to sports medicine, occupation health and military medical communities.

### Supporting information

**S1 Table. Clinical diagnostics, laboratory tests associated with heat stroke diagnosis and medical evaluation.**
(PDF)

**S2 Table. Serum electrolytes/metabolites–mean laboratory values and percent of patient relative to reference range, 0–2 days post EHS.**
(PDF)

**S3 Table. Abnormal urinalysis findings in patients, during 14 days of EHS follow-up.**
(PDF)

**S4 Table. Liver function/muscle Injury—Median laboratory values in patients, during 14 days of EHS follow-up.**
(PDF)

**S5 Table. Blood counts & coagulation—Mean laboratory values and percent of patient-values relative to reference range, during 14 days of EHS follow-up.**
(PDF)

## Author Contributions

**Conceptualization:** Matthew D. Ward, Charles Gabrial, Lisa R. Leon.

**Data curation:** Matthew D. Ward, Michelle A. King, Charles Gabrial.

**Formal analysis:** Matthew D. Ward.

**Funding acquisition:** Lisa R. Leon.

**Investigation:** Matthew D. Ward, Michelle A. King, Charles Gabrial.

**Methodology:** Matthew D. Ward, Michelle A. King, Charles Gabrial, Lisa R. Leon.

**Project administration:** Matthew D. Ward, Lisa R. Leon.

**Resources:** Charles Gabrial, Lisa R. Leon.

**Supervision:** Lisa R. Leon.

**Writing – original draft:** Matthew D. Ward.

**Writing – review & editing:** Robert W. Kenefick, Lisa R. Leon.

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
