## [Decision Letter · Decision Letter 0]

3 Dec 2019

PONE-D-19-27217

Biochemical recovery from exertional heat stroke follows a 16-day time course

PLOS ONE

Dear Dr Leon,

Thank you for submitting your manuscript to PLOS ONE. After careful consideration, we feel that it has merit but does not fully meet PLOS ONE’s publication criteria as it currently stands. Therefore, we invite you to submit a revised version of the manuscript that addresses the points raised during the review process.

We would appreciate receiving your revised manuscript by Jan 17 2020 11:59PM. To enhance the reproducibility of your results, we recommend that if applicable you deposit your laboratory protocols in protocols.io, where a protocol can be assigned its own identifier (DOI) such that it can be cited independently in the future. For instructions see: http://journals.plos.org/plosone/s/submission-guidelines#loc-laboratory-protocols

We look forward to receiving your revised manuscript.

Kind regards,

Christopher James Tyler, PhD

Academic Editor

PLOS ONE

Journal Requirements:

3. Please provide more information in the methods section on how the laboratory measurements were performed.

4. Thank you for including your ethics statement: The study was reviewed by the institutional review board and determined this research exempt from written informed consent requirements.

a) Please provide an amended Funding Statement that declares *all* the funding or sources of support received during this specific study (whether external or internal to your organization) as detailed online in our guide for authors at http://journals.plos.org/plosone/s/submit-now.  

b) Please state what role the funders took in the study.  If any authors received a salary from any of your funders, please state which authors and which funder. If the funders had no role, please state: "The funders had no role in study design, data collection and analysis, decision to publish, or preparation of the manuscript."

6. Please include your tables as part of your main manuscript and remove the individual files. Please note that supplementary tables (should remain/ be uploaded) as separate "supporting information" files.

Reviewers' comments:

Reviewer's Responses to Questions

**Comments to the Author**

1. Is the manuscript technically sound, and do the data support the conclusions?

Reviewer #1: Yes

Reviewer #2: Partly

2. Has the statistical analysis been performed appropriately and rigorously? 

Reviewer #1: Yes

Reviewer #2: Yes

3. Have the authors made all data underlying the findings in their manuscript fully available?

Reviewer #1: Yes

Reviewer #2: No

4. Is the manuscript presented in an intelligible fashion and written in standard English?

Reviewer #1: Yes

Reviewer #2: Yes

5. Review Comments to the Author

Reviewer #1: The purpose of this manuscript is to document the time course recovery of clinical biomarkers following a heat stroke event in the US Armed Forces. This paper adds to the body of knowledge and will be helpful in developing a targeted and data-driven approach to heat stroke recovery and return to activity/duty/work. There are a couple of items that warrant further clarification within the manuscript.

1) Within the introduction (L26-28) the authors discuss a consensus on EHS RTP/RTD. It may be helpful to provide additional sources of consensus as it relates to sport (NATA Position Statement on Exertional Heat Illness, 2015) and occupational (if any are available) on EHS recovery.

2) L75: The authors mention pre-determined patient demographic and clinical variables within the methodology. It would be helpful to include an explanation of what these variables were at this point in the manuscript.

3) Can the authors provide a rationale for their selection of 60 days here?

4) Within the discussion (L183 - 206), there is a discussion on the interpretation of clinical values and implications for treatment and return to activity, however, more discussion is needed surrounding the implications for return to activity here. The authors reiterate their findings from a liver damage/dysfunction (L188-191) and urinalysis (L202-206), however, there is no discussion on how return to activity should be guided in these instances. Should all activity cease until these values are within normal limits? Would activity in a cool environment (as recommended based on current consensus) be appropriate? This is an important section of the discussion and should be used to provide some insight on the current clinical recommendations on return to activity and how those may (or may not align) with the findings from this study.

5) In table 2, please indicate that the amplitude of the peak is in reference to the relative fold change compared to ULL/LLN

Reviewer #2: I read this article with great interest as it serves to illuminate an area that remains poorly defined in the literature. The manuscript discusses 2,529 exertional heat stroke (EHS) events, and the observed biochemical analysis during recovery. The authors additionally note that these labs are critical in the prognosis and return to duty decisions. The authors then subsequently describe a profile of laboratory recovery concluding that the biochemical recovery spans sixteen days. This summary, while very useful in describing a large cohort, however, is somewhat misleading and fails to address previously described and well documented variation dependent upon clinical presentation and management, in the section on limitations.

The value of laboratory tests and the biochemical profile is first and foremost critical in initial management decisions for clinicians. The tone of the manuscript appears to ignore the critical role of management in the impact on the biochemical profile, almost to the point of suggesting that management differences may have little impact on the shape of the biochemical profiles's recovery. The literature is robust with articles on EHS demonstrating the key impact of time to cooling on EHS parameters at presentation, to include morbidity and mortality and the need for dialysis. The roles of aggressive hydration, intra-vascular cooling in management also have shown an impact on the biochemical profile, and clinical recovery. Accordingly, while clearly this is a value added contribution to the literature in a global description of profiles in non fatal EHS events, I strongly believe the data could be interpreted as misleading in failing to recognize the uniqueness of each EHS event with the critical role of early cooling and its impact, as well as the role of effective management. I would argue that time to cooling and management have an impact on the biochemical profile.

Several article to consider:

Varghese GM. Predictors of multi-organ dysfunction in heatstroke.

Emerg Med J. 2005;22:185-187.

Mil Med. 2009 May;174(5):496-502.

Exertional heatstroke: early recognition and outcome with aggressive combined cooling--a 12-year experience.

Sithinamsuwan P1

A Tale of Two Heat Strokes: A Comparative Case Study. Stearns RL, Casa DJ, O'Connor FG, Lopez RM. Curr Sports Med Rep. 2016 Mar-Apr;15(2):94-7

These articles and others emphasize the the most important point in that all EHS events are not the same, and management strategies can be dramatically different depending upon initial risk stratification, which will impact biochemical profiles. This concept has to be clearly recognized as a study limitation, and reality of EHS management.

Finally, the authors discuss the cost of each EHS event as $5,337 per episode. There is no clear discussion that details how this is concluded; clearly this is markedly underestimated. I do not understand role in this manuscript.

Conclusion: this is an important contribution to the literature, however, it needs to be discussed in proper context.

6. PLOS authors have the option to publish the peer review history of their article (what does this mean?). If published, this will include your full peer review and any attached files.

Reviewer #1: No

Reviewer #2: No

---

## [Author Response · Author response to Decision Letter 0]

14 Jan 2020

We thank the reviewers for their helpful insight into the manuscript, which we have revised accordingly. We provide details of the responses and edits made to the manuscript and feel this has significantly improved the manuscript overall,

---

## [Decision Letter · Decision Letter 1]

11 Feb 2020

Biochemical recovery from exertional heat stroke follows a 16-day time course

PONE-D-19-27217R1

Dear Dr. Leon,

We are pleased to inform you that your manuscript has been judged scientifically suitable for publication and will be formally accepted for publication once it complies with all outstanding technical requirements.

With kind regards,

Christopher James Tyler, PhD

Academic Editor

PLOS ONE

Additional Editor Comments (optional):

Reviewers' comments:

Reviewer's Responses to Questions

**Comments to the Author**

1. If the authors have adequately addressed your comments raised in a previous round of review and you feel that this manuscript is now acceptable for publication, you may indicate that here to bypass the “Comments to the Author” section, enter your conflict of interest statement in the “Confidential to Editor” section, and submit your "Accept" recommendation.

Reviewer #1: All comments have been addressed

Reviewer #2: All comments have been addressed

2. Is the manuscript technically sound, and do the data support the conclusions?

Reviewer #1: Yes

Reviewer #2: Yes

3. Has the statistical analysis been performed appropriately and rigorously? 

Reviewer #1: Yes

Reviewer #2: Yes

4. Have the authors made all data underlying the findings in their manuscript fully available?

Reviewer #1: Yes

Reviewer #2: Yes

5. Is the manuscript presented in an intelligible fashion and written in standard English?

Reviewer #1: Yes

Reviewer #2: Yes

6. Review Comments to the Author

Reviewer #1: The authors have addressed all concerns from the original draft. This manuscript is going to be a tremendous addition to the scientific literature.

Reviewer #2: The authors have sufficiently addressed all my concerns; this should be a welcome addition to the literature. I look forward to citing in future manuscripts.

7. PLOS authors have the option to publish the peer review history of their article (what does this mean?). If published, this will include your full peer review and any attached files.

Reviewer #1: No

Reviewer #2: No

---

## [Editor Report · Acceptance letter]

18 Feb 2020

PONE-D-19-27217R1 

Biochemical recovery from exertional heat stroke follows a 16-day time course 

Dear Dr. Leon:

I am pleased to inform you that your manuscript has been deemed suitable for publication in PLOS ONE. Congratulations! Your manuscript is now with our production department. 

With kind regards,

on behalf of

Dr. Christopher James Tyler 

Academic Editor

PLOS ONE